# The Proteomics of T-Cell and Early T-Cell Precursor (ETP) Acute Lymphocytic Leukemia: Prognostic Patterns in Adult and Pediatric-ETP ALL

**DOI:** 10.3390/cancers16244241

**Published:** 2024-12-19

**Authors:** Fieke W. Hoff, Lourdes Sriraja, Yihua Qiu, Gaye N. Jenkins, David T. Teachey, Brent Wood, Meenakshi Devidas, Shaina Shockley, Mignon L. Loh, Evangelia Petsalaki, Steven M. Kornblau, Terzah M. Horton

**Affiliations:** 1Department of Internal Medicine, UT Southwestern Medical Center, Dallas, TX 75390, USA; fiekehoff06@gmail.com; 2European Molecular Biology Laboratory, Hinxton CB10 1SD, UK; l.sriraja.12@ucl.ac.uk (L.S.); petsalaki@ebi.ac.uk (E.P.); 3Department of Leukemia, UT MD Anderson Cancer Center, Houston, TX 77030, USA; yihuaqiu@mdanderson.org (Y.Q.); sashockley@mdanderson.org (S.S.); skornblau@mdanderson.org (S.M.K.); 4Department of Pediatrics, Texas Children’s Cancer Center, Baylor College of Medicine/Dan L Duncan Cancer Center, Houston, TX 77030, USA; gnjenkin@texaschildrens.org; 5Department of Pediatrics, The Center for Childhood Cancer Research, Children’s Hospital of Philadelphia, The Perelman School of Medicine at the University of Pennsylvania, Philadelphia, PA 19104, USA; teacheyd@chop.edu; 6Department of Pathology and Laboratory Medicine, Children’s Hospital Los Angeles, Keck School of Medicine, University of Southern California, Los Angeles, CA 90027, USA; bwood@chla.usc.edu; 7Department of Global Pediatric Medicine, St. Jude Children’s Research Hospital, Memphis, TN 38105, USA; mini.devidas@stjude.org; 8Division of Hematology, Oncology, BMT, and Cellular Therapies, Seattle Children’s Hospital, Seattle, WA 98105, USA; mignon.loh@seattlechildrens.org

**Keywords:** leukemia, RPPA, RNA, ETP, T-ALL, pediatric, adult, proteomics

## Abstract

Acute lymphocytic leukemia (ALL) is the most common malignancy in children; T-ALL accounts for 15% of ALL in children and less the 1% of all new cancer cases in adults. While overall survival (OS) in children is >90% at 5 years, 5-year OS rates are <50% in adults. The early T-cell precursor (ETP) T-ALL subtype is prognostically unfavorable in adults, but less so in pediatric T-ALL, where the diagnosis and prognosis of ETP and “near”-ETP are controversial. We aimed to compare protein and RNA expression patterns in pediatric and adult T-ALL patients to identify prognostic subgroups, and to further characterize ETP and near-ETP T-ALL in both age groups. Our results show that there are prognostic protein expression patterns in both adult and pediatric ETP ALL, and that there are differences in protein expression between pediatric and adult T-ALL that may explain differences in outcome between age groups.

## 1. Introduction

T-cell acute lymphocytic leukemia (T-ALL) is an aggressive lymphoid malignancy that arises from the oncogenic transformation of the immature T-cell precursor, resulting in a clonal proliferation of the lymphoblast [1]. ALL is the most common malignancy in children, with T-ALL accounting for approximately 10–15% of pediatric ALL cases. In adults, ALL accounts for less than 1% of all new cancer cases, with T-ALL comprising ~25% of the adult ALL [2]. While overall survival (OS) of T-ALL in children is >90% at 5 years, OS in adults is inferior with 5-year OS of 50% to 57% [3]. Survival rates are particularly poor (<10%) for both children and adults with relapsed disease [4,5].

The pathogenesis of T-ALL is heterogeneous, with underlying molecular mechanisms mainly related to signaling and cell developmental pathways involved in thymocyte proliferation, differentiation, survival, and metabolism. Distinct genetic subgroups have been identified that can be matched, to some extent, to stages of differentiation. However, the prognostic implications of these genetic subgroups are still controversial [6], and most staging subgroups are not formally included in the WHO classification [7]. An exception is the poor prognostic subgroup in adults, early T-precursor (ETP)-ALL. This group is characterized by a distinct immunophenotype (typically CD1a-, CD8-, absence or dim CD5, and positive for one or more stem cell or myeloid antigens) and similar gene expression profile with alteration in genes encoding for regulators of hematopoietic stem cells [8,9]. In contrast to adult patients, where the ETP phenotype is associated with poor outcomes, the ETP phenotype is not associated with an inferior OS or event-free survival (EFS) in pediatric T-ALL [10,11]. Potentially confusing the prognostic impact of ETP status is the existence of a subgroup of patients called “near-ETP” that are transcriptionally similar to ETP but which differ in cell surface marker expression (chiefly higher CD5) [12] but which also seem to have an inferior outcome in adult-T-ALL [13] and with unknown prognostic effect in pediatric patients.

Proteins are the central effectors that drive leukemia cell function, and the assessment of alterations in protein abundance, post-translational modification (PTM), and activation states have the potential to provide information representing the summary of genetic, epigenetic, and environmental actions in the cell. Using the reverse phase protein array (RPPA) technology and RNA sequencing, we aimed to compare protein and RNA expression patterns in pediatric and adult T-ALL to identify T-ALL subgroups, and to further characterize protein expression in the adult and pediatric ETP phenotype. 

## 2. Material and Methods

### 2.1. Sample Collection

Peripheral blood samples (n = 284) and bone marrow (BM) samples (n = 77) were collected from 361 newly diagnosed patients (age < 18 years, n = 268 (74%)) and adults (age ≥ 18 years, n = 93 (26%)) with T-ALL prior to exposure to chemotherapy, and 29 normal, unstimulated, bone marrow-derived CD34+ samples obtained from healthy adult donors purchased from AllCells (Alameda, CA, USA). Written informed consent was obtained in accordance with the declaration of Helsinki and local investigational review boards as per institutional requirements, including the IRB of the pediatric Baylor College of Medicine/Dan L Duncan Cancer Center and the University of Texas MD Anderson Cancer Center (MDACC). Characteristics of the T-ALL patients enrolled in the AALL1231 were not significantly different from the overall AALL1231 cohort.

### 2.2. Evaluation of ETP Phenotype

ETP status for patients who were enrolled on the AALL1231 was centrally assessed. Patients with T-lymphoblasts that were CD8^–^ and CD1a^–^ (<5% positive), weakly expressed CD5 (either <75% positive or median intensity >1 log less than that in mature T cells), and expressed one or more myeloid or stem cell markers (>25% positive), including CD13, CD33, CD34, CD117, and HLA-DR were classified as ETP. Patients with stronger CD5 expression but otherwise met the ETP immunophenotype were classified as near-ETP. The remaining cases were defined as having a non-ETP phenotype [10].

ETP status in T-ALL adults was defined as CD8- and CD1a- (<5% positive cells) and expression, absent or dim (<75% positive cells) CD5 expression, and expression (>25% positive cells) of 1 or more myeloid (CD11b, CD13, CD33, CD117) or stem cell (CD34, HLA-DR) markers [8]. Patients with a similar phenotype that was CD5+ were defined as near-ETP. The remaining cases were defined as having a non-ETP phenotype. ETP phenotype assessment in adults was performed independently as part of routine diagnostic workup by the hematopathologists at MDACC.

### 2.3. Treatment Regimens

#### 2.3.1. Pediatric Treatment Protocol: Children’s Oncology Group (COG) AALL1231

Two-hundred ninety-two patients, including all 268 pediatric cases (age 0–17) and 24 adolescents and young adult (AYA) patients (age 18–30), were enrolled in the COG AALL1231 phase 3 clinical trial; 139 were treated on arm A (modified augmented Berlin-Frankfurt-Münster (AFBM) backbone with intrathecal cytarabine, vincristine (VCR), dexamethasone, daunorubicin, PEG-asparaginase, and intrathecal methotrexate), and 150 on arm B (modified AFBM backbone plus bortezomib in induction and delayed intensification) [14]. Three patients in our patient cohort were ineligible and excluded from the clinical trial.

#### 2.3.2. Adult Treatment Protocols

The 69 adult T-ALL patients were treated under a variety of protocols, and 13 were treated off protocol. Ten were treated according to an AFBM protocol, 53 received hyper-CVAD (cyclophosphamide (CTX), VCR, doxorubicin, dexamethasone (VAD)), alone (n = 23), in combination with nelarabine (n = 27) or augmented hyper-CVAD (n = 3). A single patient each received mini-HCVD plus venetoclax; VAD + nelarabine; the combination of clofarabine, etoposide, CTX, VCR, and bortezomib; or cladribine plus sorafenib (Appendix A).

### 2.4. Sample Processing

As previously described [15,16,17], pediatric leukemia cell-enriched fractions were generated remove “by” using lymphoprep solution (Axis-Shield, Oslo, Norway), followed by T-cell isolation using the MACSxpress^®^ Pan T Cell Isolation Kit (Miltenyi Biotec, Cologne, Germany). Whole-cell protein lysates were made from sorted mononuclear cell fractions and normalized to a concentration of 10,000 cells/µL. Adult samples were enriched for leukemic cells using Ficoll separation or lymphoprep solution, and the resulting cells were made into whole cell lysates.

### 2.5. RPPA Methodology

Samples were printed in five serial dilutions onto slides along with normalization and expression controls. Control samples included both a negative (Laemmli sample buffer, Bio-Rad Laboratories Inc., Hercules, CA, USA) and positive controls (a mixture of 11 different cell lines known to express the proteins of interest). Slides were printed, and samples were probed with 321 strictly validated primary antibodies, including 251 antibodies directed at total proteins and 70 targeting post-translational modified (PTM) proteins (cleaved form (n = 4), histone 3 methylation sites (n = 6), phosphorylation (n = 60)). Stained slides were analyzed using Microvigene software (Version 3.4, Vigene Tech, Carlisle, MA, USA). Single-value log2 protein concentrations were generated from the five serial dilutions using the SuperCurve algorithm [18]. Loading control [19] and topographical normalization [20] procedures were performed to account for protein concentration and background staining variations. Protein expression levels were adjusted relative to the median of the 29 normal CD34+ BM samples.

### 2.6. RNA-Sequencing

RNA was isolated from 81 T-ALL samples to prepare libraries using the TruSeq RNA Exome RNA Kit (Illumina, San Diego, CA, USA). The libraries were sequenced on an Illumina NextSeq 500 (n = 65) or NovaSeq 6000 (n = 16) using 150-bp paired-end chemistry at a target depth of 40–50 M fragments per sample. FastQC (v.0.11.9) (https://github.com/s-andrews/FastQC) with default parameters was used for quality control, and samples with Phred quality scores greater than 20 were selected. Reads were aligned to the human genome (GRCh38) using HISAT2 (v2.1.0) with default settings [21]. Gene count analysis was performed using HTseq-count [22], and Ensembl gene IDs were converted to HGNC symbols using the R package biomaRt (v2.54.0). Batch correction was applied to the count data using ComBat-Seq from the sva package (version 3.50.0) (Appendix A) [23]. One sample appeared as an outlier in the COMBAT-corrected counts used for transcription factor activity inference (Appendix A). Because there was no indication in the metadata or the fastQC quality control metrics to justify excluding the sample, and the variance stabilizing transformation-normalized counts did not reveal any outliers (Appendix A), we have included this sample in the final analysis.

### 2.7. “MetaGalaxy” Analysis

The “MetaGalaxy” (MG) analysis was performed as previously published [17]. The rationale behind this approach was that while the traditional unsupervised clustering takes all patients and proteins together, weighting all equally, the MG approach allocates proteins into smaller “Protein Functional Groups” (PFG). Briefly, each individual protein (n = 321) was allocated into one or more PFG based on their known (functional) relationships or strong correlations within the dataset. The progeny clustering algorithm [24] was applied to each PFG to cluster patients into “Protein Clusters” (PC), a subset of patients with similar a correlated relative protein expression pattern. A binary matrix was built for each patient and each PC (i.e., “1” if a patient is a member of a PC within a PFG and “0” if a patient is not a member of that PC). Co-clustering of the binary matrix [25] was used for simultaneous clustering of the rows (i.e., PC) and columns (i.e., patients), resulting in the identification of higher-order structures between those PC. The strong correlation between various PC was defined as a “Protein Constellation” (CON). Patients who expressed similar patterns of CON were defined as a “Protein Expression Signature” (SIG).

### 2.8. Statistical Analysis

Continuous and categorical variables were summarized by reporting medians and ranges or frequencies and percentages, respectively. Estimates of OS, EFS, and complete remission (CR) duration were calculated using the Kaplan-Meier method. OS and EFS were defined as the time from diagnosis until death or until relapse or death, respectively. CR duration was defined as the time from CR to relapse or death. Group differences for censored outcomes were calculated using the log-rank test.

To identify subsets of adult patients with a high- and lower relapse risk among patients treated on adult treatment protocols evaluable for remission duration (n = 54), patients were randomly divided into a training (60%) and test (40%) set. Univariate Cox regression analysis was applied to identify proteins prognostic of relapse. Proteins with a *p*-value < 0.01 were selected for the multivariate Cox regression model via penalized maximum likelihood [26]. Based on the selected proteins, a risk score was constructed using the following equation:$$Risk\;score\;=\;\sum \beta_{protein}\;Exp_{protein}$$
where $\beta_{protein}$ represents the multivariate cox regression coefficient of each individual protein and $Exp_{protein}$ represents the expression level of each protein. Patients were stratified into low- and high-risk clusters based on a cut-off of the median risk score. The receiver operating curve (ROC) was calculated using SurvivalROC (Version 1.0.3.1) to assess the predictive accuracy of the risk score for the training and test sets [27].

Gaussian mixture model clustering was applied to the near/ETP subset of the RPPA data using mClust [28] to identify different subsets of ETP phenotype patients. To identify different patient subsets among the total cohort of patients, UMAP clustering analysis [29] was applied to the RPPA protein data, as well as to the top 5% variable genes for patients with available RNA sequencing data. The optimal number of clusters was selected using the Bayesian information criterion (BIC). Differential expression analyses were performed using limma [30] for the RPPA protein data, DESeq2 [31] for the RNA count data, and gene enrichment analyses for the RNA cluster’s Gene Ontology (GO) and Molecular function enrichment. Transcription factor activity was inferred using decoupleR for the differentially expressed genes between subgroups [32]. All statistical tests and plots were generated in R (Version 1.3.959—2009–2020 Rstudio, Inc., Boston, MA, USA). Protein networks were generated in Cytoscape (Version 3.8.0) [33,34].

## 3. Results

### 3.1. T-ALL Cells Express Recurrent Patterns of Relative Protein Expression Levels That Classify T-ALL Patients into a Finite Number of Protein Expression Signatures

Protein expression levels for all 321 antibodies were calculated relative to the median of the nonmalignant CD34+ cells. Based on their known functionality from the literature or strong correlation in the T-ALL data set, proteins were allocated into 32 protein functional groups (PFG) (e.g., cell cycle, apoptosis) (Appendix A). A clustering algorithm was applied to each PFG to identify recurrent relative protein expression patterns in T-ALL and identified subgroups of patients with a similar correlated pattern of relative protein expression levels defined as a “Protein Cluster” (PC) (range, 3–5 PC within a PFG). A total of 125 PC for all 32 PFGs were identified (Appendix A). PC was not specific for age, with the frequency of pediatric or adult cases within the expected number of two standard deviations from the median (Appendix A). Heatmaps for each PFG are published online: http://leukemiaproteinatlas.org/pediatric-all/).

To define an even more integrated system-wide vision of protein expression patterns in T-ALL, PC-membership for each patient was combined into a large binary matrix (125 rows × 361 columns). Co-clustering analysis (i.e., simultaneous clustering of PCs (rows) and patients (columns)) was applied to search for correlation between PC-membership from various PFGs across the 361 patients enabling recognition of 13 patterns of strongly correlated PCs from various PFGs, defined as a “Protein Constellation” (CON) Figure 1, *y*-axis horizontally, C1 to C13 Appendix A. Patients with a similar pattern of CONs were grouped as a “Protein Expression Signature” (SIG) (n = 10) (Figure 1, *x*-axis, vertically, S1 to S10).

### 3.2. Protein Expression Signatures Are Associated Disease and Patient Characteristics

Clinical characteristics and disease features were correlated with SIGs (Table 1, annotation bar Figure 1). SIG-membership was highly significantly associated with age group (adult vs. pediatric) and age subclass (*p* < 0.001), although there were no signatures exclusively associated with a single age subclass. Infant (age 0–1), pediatric (age 2–17 years), and AYA (age 18–30) cases exclusively comprised SIG-2, 3, and 6, which had no adults over the age of 30 years, and adults (11% overall) only comprised 3% and 4% respectively of SIG-1 and 7. Patients aged ≤ 1 year (4% overall) were somewhat overrepresented in SIG-1, 6, and 9 (9%, 8%, and 7%, respectively) and absent from SIG-2, 4, and 5, and adults (11% overall) and AYA (15% overall) cases dominated SIG-5 (70%), and SIG-10 (55%). In contrast to the age-biased SIGs, there were three SIGs (4, 8, and 9) with a proportional mixed composition across all age groups.

ETP (14% overall) and near-ETP (15% overall) phenotypes were seen in all SIGs except SIG-6, with ETP enriched in SIG-4 (27%) and SIG-10 (32%) (*p* = 0.016). Near-ETP was fairly proportional across the other nine SIGs. The combination of either ETP or near-ETP (29% overall) was underrepresented in SIG-1 (16%) and SIG-2 (20%) and overrepresented in SIG-4 (36%), SIG-9 (37%) and SIG-10 (53%) (*p* = 0.002). None of the SIGs was ETP-specific. Patients with central nervous system (CNS) involvement were overrepresented in SIG-7 and 8. CON-9 (*p* = 0.016), as was higher white blood cell count (*p* < 0.001). No associations were seen for gender, ethnicity, and chromosomal translocations involving the T-cell receptor.

### 3.3. Protein Expression Signatures Are Not Prognostic for Outcome

To minimize the effect of treatment approaches, survival analysis was performed for the patients treated on the COG protocol and adult treatment protocols separately. Patients treated with the COG AALL1231 had a median follow-up of 173 weeks, with 86% (n = 249) of the patients still alive at the end of the follow-up. Adult patients had a median follow-up of 71 weeks, with 27 (39%) patients that were still alive at the end of follow-up. SIGs were not prognostic for clinical outcome among patients in either cohort (COG AALL1231 OS: *p* = 0.90, EFS: *p* = 0.82; adult treatment protocols: OS: *p* = 0.62, CR duration *p* = 0.67) (Appendix A).

### 3.4. Prognostic Risk Score Stratifies Adult T-ALL Patients into Low- and High-Risk Protein Clusters

Next, we wanted to assess if we could distinguish patients with lower and higher relapse risk based on their protein expression profiles. Because studies have demonstrated that pediatric-inspired regimens improve survival for AYA and to minimize the effects of different treatment approaches confounding the analysis, AYA patients treated on the pediatric COG protocol were excluded from the analysis, resulting in a total of 53 adult patients with known evaluable data for CR duration analysis.

Univariable analysis followed by multivariable Lasso Cox regression analyses were applied to the training set and identified 15 proteins that significantly contributed to CR duration (Figure 2A) with high expression of TGM2, EIF4G2, FOXO3A, and BCL2L1 and phosphorylated CDKN1B.pS10 being associated with higher risk, whilst high expression of PTEN, CCND1, and XPA were three of the protein markers associated with low took out remission risk. Using the regression coefficients of the generalized linear model, a risk score was calculated for each individual patient to stratify patients into a low-risk protein cluster (LR-PC) and a high-risk protein cluster (HR-PC) based on the median cut-off value of the risk score among all adult patients (Figure 2B). Time-dependent receiver operator characteristic (ROC) analysis assessed the predictive performance of the risk score, yielding an area under the curve (AUC) of 0.97 (Figure 2C). The robustness of the risk groups based on the risk score was validated in the test set with an AUC of 0.77 (Appendix A). OS and CR duration were significantly associated with the risk-assigned protein clusters (OS, LR-PC: median not reached (NR) vs. HR-PC: median 106 weeks, *p* < 0.001; CR duration, LR-PC: median NR vs. HR-PC: median 57 weeks, *p* < 0.001, respectively) (Figure 2D). Clinical variables did not differ between the two risk protein clusters, including age, white blood cell (WBC) count, sex, ethnicity, and ETP phenotype.

### 3.5. Protein Clustering Analysis Identifies a Subset of Pediatric ETP Phenotype Patients Associated with Inferior Clinical Outcome

In adults, the ETP-phenotype ALL is associated with a higher mutational burden, more resistance to treatment, and inferior outcome compared to non-ETP T-ALL [8], however, in pediatric patients, the ETP impact may not be prognostically relevant [10]. As noted above, the existence of near-ETP cases confuses the prognostic impact of ETP and near-ETP status, but in both the adult and pediatric sets, ETP and near-ETP cases had similar outcomes (Appendix A). We asked whether ETP and near-ETP shared similar protein expression profiles and whether proteomics might define subsets with adverse prognoses. Gaussian mixture model clustering of all the ETP and near-ETP patients was performed, and the Bayesian information criterion score was used to determine the optimal number of three [take out-clusters] ETP clusters (Appendix A).

ETP clusters were clearly associated with age groups (*p* < 0.001), with one cluster that was dominated by adult ETP patients (ETP-A) (navy blue; n = 15/20), one cluster that included mostly pediatric patients (ETP-P) (pink; n = 13/15) and one protein cluster that was comprised of both adult and pediatric patients (ETP-MX) (green; pediatric (n = 45/68), adult (n = 23/68)) (Appendix A). ETP and near-ETP cases were found proportionally distributed in all three clusters and had highly similar protein expression patterns, suggesting that they are very similar at the proteomic level. Furthermore, ETP vs. near-ETP cases had similar outcomes in the cluster (Appendix A). Based on relatively similar protein levels, as well as OS between near-ETP and ETP patients, ETP and near-ETP were combined for the analysis (n = 103). Model-based clustering using finite mixture models was applied, and differential expression analysis was performed for each cluster separately compared to non-ETP T-ALL samples, which identified 14 proteins that most strongly discriminated between the three clusters (Figure 3A). PI3K/Akt signaling proteins (AKT1, AKT3 IGF1R, and PIK3CB) had the highest relative expression among patients in ETP-P, SMAD5-pS463_465, and RICTOR-pT1135 were high both in ETP-A and ETP-P. Histone 3 expression was highest among patients in ETP-MX. The clinical outcome of pediatric/AYA patients treated on the AALL1231 trial with an ETP-P protein profile (Figure 3B, pink) was inferior compared to the pediatric/AYA patients in the ETP-MX (Figure 3B, light green), which had similar outcome to the non-ETP T-ALL counterpart (Figure 3B, red). Clinical characteristics were similar between the ETP-P and ETP-MX patients, except for WBC count, which was lower among patients that fell in the ETP-P (Appendix A). Among adult patients, the ETP clusters were not prognostic (Appendix A).

### 3.6. Cluster Analysis Reveals Pediatric T-ALL Subset Enriched for Upregulation of Various RNA Signaling Pathways

Next, we wanted to identify RNA-sequencing clusters and compare these to the RPPA protein clusters. RNA-sequencing data was available to 81 patients enrolled in the AALL1231 COG trial. Unsupervised clustering identified four RNA clusters (RC) (Figure 4A). RNA clustering was markedly different from the RPPA protein clustering, with 56% of the 81 samples falling in a different cluster (Fisher’s Exact test, *p* = 0.114) (Figure 4B). For each RC, several up- or down-regulated genes were found to be discriminatory. As an example, RC-1 was enriched for *COL1A1* and *COL6A1, LITD1* and *CFTR* were upregulated in RC-1, whilst *GJA1* was notably upregulated in RC-3. RC-4 exhibited upregulation of several genes, including *TRPC3*, *IL-4*, and *RPS6KA6*.

Based on the identified gene set enrichments in each cluster, we could recognize three distinct patterns associated with the four clusters. Cluster 1 (salmon red) and Cluster 4 (purple) were enriched for several gene sets related to integrin pathways and interleukins, while Cluster 2 (green) was mostly depleted for these. Cluster 3 (blue) was negatively associated with pathways involved in cell cycle and kinase signaling (e.g., *ATR, E2F, FOXM1, PLK1*), while these gene sets were mostly enriched in Cluster 2, 3 and 4 (Figure 4C).

### 3.7. ETP T-ALL Associated with Apoptosis and TP53 Pathway RNA Gene Expression

To further investigate the ETP phenotype at the transcriptome level, we combined ETP and near-ETP samples (n = 15) and compared them to non-ETP samples (n = 66) using differential gene expression analysis. A total of 1539 differentially expressed genes (FDR-adjusted *p*-value < 0.05) were identified (Figure 5A). Unsupervised clustering of the most differentially expressed genes (log2FC ≥ 1.5) did not separate ETP from non-ETP patient samples (Figure 5B). To summarize this information, transcription factor interference was performed using a statistical method that utilized a meta-resource comprising over 100 databases of prior knowledge [35]. Genes upregulated in ETP/near-ETP compared to non-ETP were involved in the TP53 pathway, apoptosis, and unfolded protein response, as well as the *MYC* pathway and *mTOR*. Non-ETP was associated with *KRAS* and several cell cycle and cell division pathways: E2F targets, G2M checkpoint, and mitotic spindle (Figure 5C). Transcription factor activity inference revealed that *SMAD4* and *MYC* had upregulated activity in the ETP/near-ETP group, whereas *E2F* and *TCF7* transcription factors were downregulated (Figure 5D).

## 4. Discussion

In this study, we used RPPA methodology to measure relative protein expression levels in a cohort of pediatric and adult patients with T-ALL. Similar to what has been published previously by our group in other leukemia types, we identified recurrent protein expression signatures across all T-ALL patients. However, unlike AML, we were unable to determine specific SIGs associated with the outcome. We also examined protein expression among ETP patients and noted that ETP-T-ALL pediatric patients with certain protein expression patterns did poorly compared to other pediatric ETP groups. Lastly, the clustering of adult T-ALL patients identified a set of proteins associated with increased relapse risk that enabled stratification into two risk groups.

The protein landscape of T-ALL is heterogeneous due to a complex interplay of different protein expression levels and activation states in a subset of patients. The presence of recurrent protein expression signatures within pediatric and adult T-ALL, however, suggests that certain protein dependencies are likely shared across patient subgroups, whereas other protein expression pathways are more subgroup-specific. Similarly, the uneven distribution of age groups among the different SIGs, without any age-exclusive subgroups, suggests that adult and pediatric patients may rely on similar protein pathways, though some pathways are more strongly associated with certain age groups. This could potentially contribute to differences in treatment resistance patterns between age groups. A summary of all deregulated proteins relatively to normal that could guide the process of target selection is shown in Appendix A.

Furthermore, we identified a set of proteins associated with a higher risk of relapse risk in adult T-ALL patients, a challenging group to treat with poor prognostic outcomes, particularly following relapse. The proteins with a differential expression included CDKN1B-pS10, BCL2L1, EIF4G2, FOXO3A, PTEN, and CCND1, which are primarily involved in cell cycle regulation, proliferation, and protein synthesis. This suggests that these biological processes are likely linked to prognosis. The predictive accuracy of this signature was validated on a test subset from the same cohort of adult T-ALL, demonstrating that the prognostic risk score reliably predicted OS and EFS outcomes (Appendix A).

In addition to the relationship between SIGs and age, we also observed an association with ETP phenotype, although the SIGs were not specific to ETP. Given that the prognosis of ETP remains similar to the general T-ALL pediatric population, unlike in adults, where it is linked to poor outcomes, we further explored protein patterns in both near-ETP and ETP T-ALL. Our data suggests that ETP and near-ETP exhibited highly similar protein expression, where they were largely indistinguishable and did not differ in outcome. Unsupervised clustering analysis identified three ETP protein clusters: one dominated by pediatric patients, one enriched for adults, and one that was age independent. A subset of 14 selected proteins was able to distinguish between these clusters, with all but two (HIST3H3, SMAD2-pS465-467) more highly expressed in the ETP-P patients. Notably, pediatric patients from the ETP-P protein cluster had significantly worse OS and EFS compared to pediatric patients in ETP-MX. The outcome was also inferior compared to the non-ETP pediatric patients, even though a difference in EFS between ETP and non-ETP was not noticed in the COG AALL1231 clinical trial. Thus, we were able to define an unfavorable prognostic pediatric ETP group that can possibly account for the occasional observation that ETP does poorly. A similar observation was not seen for adults.

Previous studies have highlighted heterogeneity within ETP based on lineage plasticity [36,37]. To further explore this, we compared RNA expression profiles between ETP/near-ETP and non-ETP in a subset of pediatric T-ALL patients. Clustering identified four distinct RNA expression clusters, which, consistent with previous findings [38,39,40], showed poor correlation with the RPPA protein clusters. RNA clusters were associated with up and downregulation of gene sets, with most genes playing a role in cell cycle regulation, proliferation, cell death, or cell signaling.

This study has several limitations, including the small number of adult T-ALL patients who were treated with different treatment protocols and the limited number of 321 protein antibodies used in the RPPA analysis. Additionally, the retrospective nature of the study calls for future prospective studies to confirm the predictive value of the proteins associated with higher- and lower-risk patients.

## 5. Conclusions

This study is the first to apply RPPA protein profiling to a cohort of newly diagnosed T-ALL patients. We identified recurring protein expression patterns in T-ALL and demonstrated the ability to pinpoint a subset of adult patients with high risk for relapse, as well as a group of pediatric ETP T-ALL patients with poorer prognosis based on protein expression data. The observation that there are age-specific patterns supports the idea that the origin of T-ALL in most pediatric and adult patients is different, while overlapping patterns suggests that there are some with a common pathophysiology. Further research is needed to explore proteins in high-risk patient groups with the aim of identifying potential prognostic proteins that can aid in risk stratification.

## Figures and Tables

**Figure 1 cancers-16-04241-f001:**
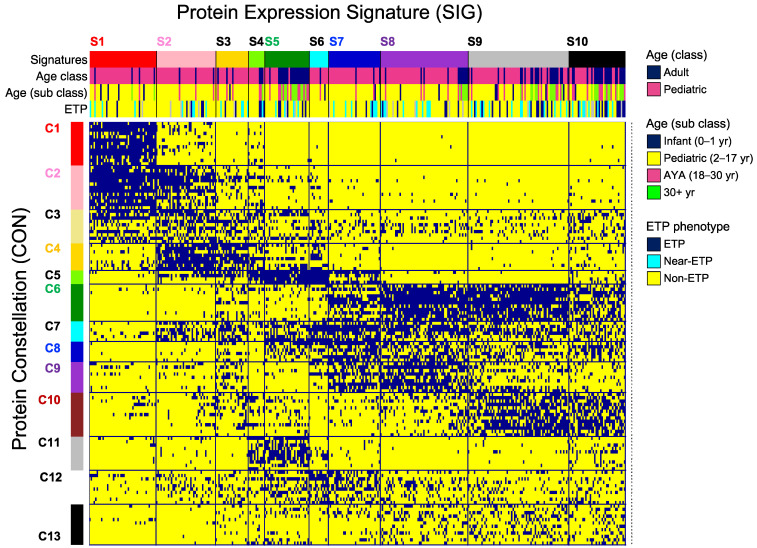
Meta-Galaxy analysis. A large binary matrix representing the 361 patients (columns) and 125 protein clusters (PC) (rows, Appendix A). A blue pixel indicates the presence of a given PC in that patient. Block clustering analysis identified the presence of 13 protein constellations (horizontally) that formed 10 protein expression signatures (vertically). Annotations are included at the top for each individual patient. Age (class): adult (navy blue), pediatric (pink); age (subclass): age (≤1 year, navy blue), pediatric (2–17 years, yellow), adolescents and young adults (pink), adult (30+ years, green); ETP phenotype: ETP (navy blue), near-ETP (light blue), non-ETP (yellow).

**Figure 2 cancers-16-04241-f002:**
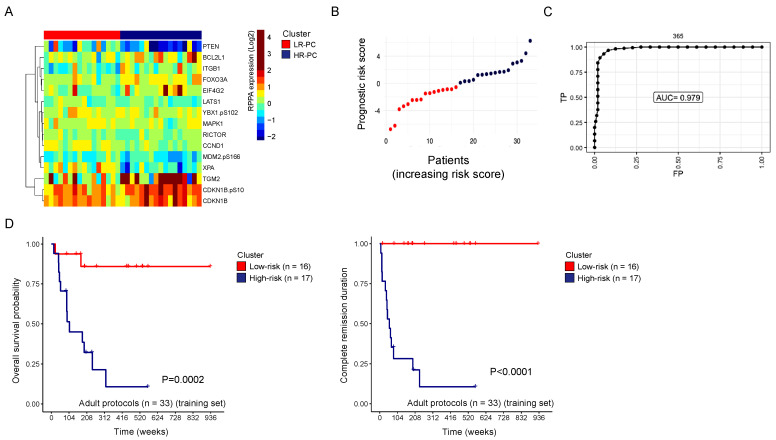
Regression analysis identified 2 protein clusters associated with clinical outcomes in adult T-ALL in the training set (n = 33). (**A**) Heatmap showing relative protein expression levels of 15 proteins significantly associated with complete remission (CR) duration in the training set (n = 33). (**B**) Prognostic risk score calculated for each individual patient. Patients were divided into a low-risk protein cluster (LR-PC) and a high-risk protein cluster (HR-PC) based on the median risk score. (**C**) Time-dependent receiver operator characteristic (ROC) analysis. (**D**) Overall survival and CR duration stratified for patients in the low-risk (red) and the high-risk (navy blue) protein cluster.

**Figure 3 cancers-16-04241-f003:**
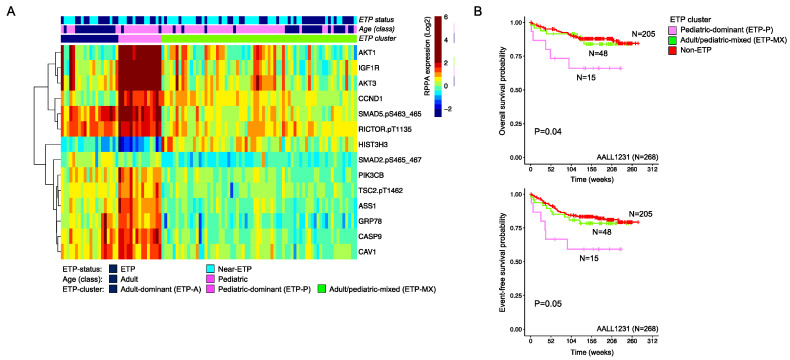
(**A**) Heatmap showing 14 proteins that were most strongly associated with the three protein clusters based on differential expression analysis. Protein expression levels are shown relative to normal, ranging from low (blue) to normal (green) to high (red). (**B**) Overall survival (upper) and event-free survival (lower) analysis for the AALL1231 patients that clustered into the pediatric-dominant (ETP-P; pink) and the adult/pediatric mixed protein clusters (ETP-MX; green). Non-ETP patients (red) were shown as a reference.

**Figure 4 cancers-16-04241-f004:**
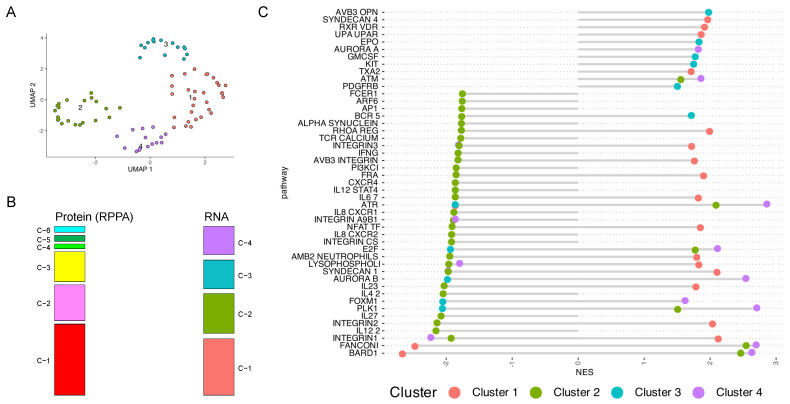
(**A**) RNA UMAP cluster analysis assignments of pediatric transcriptome data (n = 81). (**B**) Sankey plot of RPPA protein UMAP clusters and RNA-Seq UMAP clusters (**C**) Cluster-specific enrichment scores for the top representative pathways using Hallmark gene sets of upregulated and downregulated genes. The *x*-axis represents the normalized enrichment score (NES) from the FGSEA analysis, where NES < 0 indicates downregulated pathways, and NES > 0 indicates upregulated pathways.

**Figure 5 cancers-16-04241-f005:**
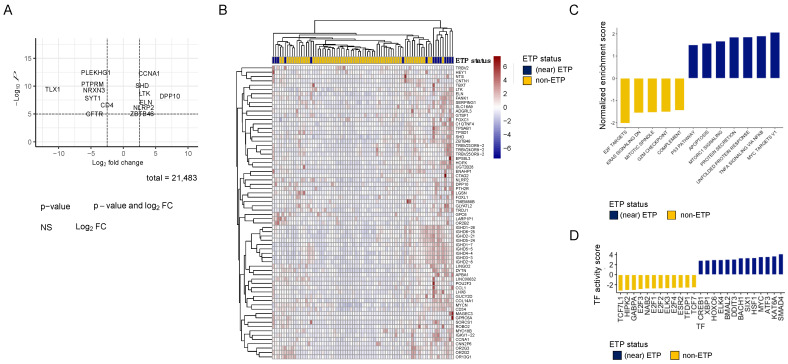
Gene expression profiling comparing near-ETP/ETP to non-ETP T-ALL. (**A**) Volcano plot displaying differentially expressed genes between near-ETP/ETP and non-ETP patient samples. Genes shown in red have an FDR *p*-value < 0.05 and an absolute log2 fold change greater than 2.5, blue points indicate genes with an FDR *p*-value < 0.05 and an absolute log2 change less than 2.5, and green and grey points represent non-significant genes. (**B**) Heatmap showing differential gene expression with a log2FC cut-off greater than 1.5. fold difference between variance stabilizing transformation normalized RNA expression counts in near-ETP/ETP and non-ETP T-ALL. (**C**) The FGSEA enrichment scores for pathways upregulated in the near-ETP/ETP are shown in blue, while those shown in yellow are down-regulated. (**D**) Transcription factor activities and transcription factors between near-ETP/ETP and non-ETP. TF activity estimates are based on the normalized gene counts (*y*-axis).

**Table 1 cancers-16-04241-t001:** Patient characteristics (N = 361).

Variable Category		Count	Freq	S1	S2	S3	S4	S5	S6	S7	S8	S9	S10	*p*
Signature	Count	361	100%	12%	11%	6%	3%	8%	4%	10%	16%	19%	11%	
Gender	Female	84	23%	29%	25%	9%	45%	30%	8%	17%	25%	21%	24%	0.345
Age (class)	Adult	93	26%	9%	8%	14%	27%	70%	8%	9%	19%	34%	55%	<0.001
	Pedi	268	74%	91%	93%	86%	73%	30%	92%	91%	81%	66%	45%	
Age (continuous)	Mean	15.2		10.7	10.7	11.2	13.4	24.9	9.5	9.9	13.5	17.5	25.9	<0.001
	SD	13.3		7.2	5.4	5.7	9.4	15.9	6.5	9.7	13.0	14.3	18.4	
Age (sub class)	≤1 years	15	4%	9%	0%	5%	0%	0%	8%	3%	3%	7%	3%	<0.001
	2–18 years	253	70%	82%	93%	82%	73%	30%	85%	89%	78%	59%	42%	
	AYA	53	15%	4%	8%	14%	18%	37%	8%	6%	10%	19%	26%	
	30+ years	40	11%	4%	0%	0%	9%	33%	0%	3%	8%	15%	29%	
Ethnicity	Hispanic or Latino	78	22%	27%	18%	18%	27%	10%	8%	4%	27%	26%	24%	0.356
	Not Hispanic or Latino	261	72%	67%	80%	77%	73%	87%	92%	77%	64%	62%	76%	
	Unknown	22	6%	7%	3%	5%	0%	3%	0%	9%	8%	12%	0%	
Race	Asian	18	5%	2%	3%	5%	0%	7%	23%	11%	5%	2%	5%	0.185
	Black or African American	33	9%	11%	3%	5%	9%	13%	15%	11%	7%	13%	5%	
	White	264	73%	78%	83%	77%	73%	77%	54%	66%	75%	65%	79%	
	Other	9	2%	0%	0%	0%	0%	0%	8%	0%	7%	4%	3%	
	Unknown	37	10%	9%	13%	14%	18%	0%	0%	11%	7%	16%	8%	
CNS-involvement ^†^	Positive	104	29%	36%	33%	18%	0%	3%	23%	40%	41%	25%	32%	0.016
	Negative	245	68%	62%	68%	82%	91%	77%	69%	60%	59%	74%	63%	
	Unknown	12	3%	2%	0%	0%	9%	20%	8%	0%	0%	1%	5%	
MRD (end of induction II) *	<0.01%	171	59%	66%	63%	50%	75%	60%	92%	56%	52%	55%	45%	0.455
	0.01–0.099%	20	7%	2%	5%	9%	0%	20%	0%	9%	8%	6%	15%	
	0.1–0.99%	23	8%	5%	5%	14%	0%	10%	0%	9%	10%	12%	5%	
	>1.0%	65	22%	18%	28%	27%	13%	0%	0%	21%	31%	20%	30%	
	Unknown	13	4%	9%	0%	0%	13%	10%	8%	6%	0%	6%	5%	
KMT2A-rearrangement	Yes	33	12%	4%	10%	9%	18%	13%	8%	3%	14%	9%	8%	0.743
	No	284	79%	76%	68%	77%	64%	80%	77%	91%	80%	82%	79%	
	Unknown	44	12%	20%	23%	14%	18%	7%	15%	6%	7%	9%	13%	
ETP phenotype	Positive	49	14%	7%	10%	9%	27%	17%	0%	9%	8%	18%	32%	0.016
	Near	54	15%	9%	10%	18%	9%	13%	0%	20%	15%	19%	21%	
	Negative	239	66%	82%	70%	68%	64%	63%	100%	69%	68%	62%	37%	
	Unknown	19	5%	2%	10%	5%	0%	7%	0%	3%	8%	1%	11%	
Risk stratification (AALL1231)	Standard risk	87	24%	33%	33%	32%	27%	10%	62%	23%	19%	24%	8%	0.490
	Intermediate risk	173	48%	53%	63%	64%	27%	20%	31%	63%	61%	35%	39%	
	Very high risk	15	4%	2%	5%	5%	0%	0%	0%	6%	8%	4%	3%	
	Unknown	86	24%	11%	0%	0%	45%	70%	8%	9%	12%	37%	50%	
WBC (continuous)	Mean	151.9		148.3	111.9	122.7	68.0	59.1	82.0	200.7	251.9	157.0	120.8	0.000
	SD	180.1		157.1	123.2	175.2	110.8	149.4	64.4	227.5	206.9	172.7	169.4	

* Only reported in patients that participated in the AALL1231 clinical trial; ^†^ CNS status: positive, CNS-2 or CNS-3; negative, CNS-1.

## Data Availability

The data presented in this study and all the analyses that were performed, including those not discussed in this paper, are published online at http://leukemiaproteinatlas.org/t-all/. The original data presented in the study is also included in the Appendix A and are available upon reasonable request (skornblau@mdanderson.org).

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
