# Peer review of "The Proteomics of T-Cell and Early T-Cell Precursor (ETP) Acute Lymphocytic Leukemia: Prognostic Patterns in Adult and Pediatric-ETP ALL"

_cancers, 2024, doi:10.3390/cancers16244241_

Round 1

Reviewer 1 Report

Comments and Suggestions for Authors The manuscript titled "The proteomics of T-ALL and Early T-cell Precursor ALL in pediatric and adult populations" by H​off et al performed reverse phase protein array (RPPA) (n=361 samples and n= 321 antibodies) coupled with RNA-sequencing (n=81) in pediatric and adult T-cell Acute lymphcytic leukemia (T-ALL) to identify prognostic subgroups and characterize Early-T-cell precursor (ETP) and near-ETPs between the two age groups. Overall the manuscript is good, however there are a few suggestions and recommendations below:   1. Section 2.2 "Evaluation of ETP phenotype": This section is a bit confusing to read at first. I recommend having a small graphical representation of ETP levels (a horizontal line), arrow up denoting adults and down for pediatric, and along the horizontal line you can define classifications.  This could even be a supplemental figure.  2. Section 2.6: First, the order should be corrected. The Batch correction should be written after the genome alignment using HISAT2. The "hisat2" should be written as HISAT2. Further, define the parameters such as batch and group in ComBat-Seq. The authors should show a PCA plot before and after batch correction to ensure proper correction.  3. The quality of RNA-seq data: Authors should comment on the quality measures used, e.g. phred quality scores. Also, how were the adapters trimmed? Please cite the appropriate tool used for the same.  4. Line 165 "rows(i.e., patients) and columns (i.e., PC)" should be corrected with respect to line 218 and Figure 1 legend (line 226) 5. Confusing axis mentions: line 221and 222. The mention of horizontal and vertical for C1 to C13 and S1 to S10 seems reversed. Please correct it accordingly.  6. For reproducibility: All the raw FASTQ files along with raw and Batch corrected HTSeq counts and RPPA results should be deposited at the NCBI - Sequence read archive (SRA).      Comments on the Quality of English Language

The data availability statement: I have with experience, do not recommend personal websites that tend to go down in couple of years. I rather have the data uploaded to NCBI. 

Reviewer 2 Report

Comments and Suggestions for Authors

The authors have conducted an important and interesting study. However, I have concerns about their inferences and data analysis. Results have been presented vaguely and can be significantly improved upon. Please see comments below.  

  1. "Conclusion. Our study suggests that there are similarities and differences in protein dependencies in pediatric, adults, and ETP subtypes 45

of T-ALL based on their protein expression patterns" While this kind of a statement sounds fine for the simple summary, the main abstract should shed some more insight into the study and conclude results in a more detailed and scientific manner.  

  1. Line 64: This group is characterized by a distinct immunophenotype and similar gene expression profile [8,9]. This sentence sounds vague and could have detailed information on immunophenotype and gene expressions.

  1. Were the authors directly involved in assessment of the ETP phenotype or was this conducted by a core facility? This should be mentioned. If the authors have access to immunophenotyping data, sample flow cytometry plots should be included as supplemental data.
  2. The authors should avoid using abbreviations in title. See Results title 3.1 and replace SIGs with signatures.
    5. Figure 1 Meta Galaxy analysis should be strengthened by adding in some information about the SIGs. Molecular information distinguishing these SIGs, and helping with ETP classification should be stated. 
  3. The manuscript repeatedly mentions "We identified recurring protein expression" in the Abstract and Conclusion. This sounds vague and fails to provide information on which protein groups, molecularly and functionally what are the characteristics of these proteins. This information should be added to strengthen the manuscript. 
  4. Based off SIGs identified in pediatric, adults, ETP subgroups, can the authors speculate distinct potential therapeutic targets?

Round 2

Reviewer 1 Report

Comments and Suggestions for Authors

The authors have incorporated all the suggestions. However, there are minor comments that should be incorporated/addressed. 

1. The batch correction figure is good, but the PC1 with 62% variance is high and have the authors corrected for high variance? What factors drive these PC? (This should be noted and mentioned in the manuscript). 

2. The data availability as RAW files (FASTQ) and RRPA should be made available or include the statement under data availability that it can be provided upon contacting the corresponding author. Unless it will reveal the identity of the subjects or other ethical conserns. 
